

# The endoplasmic reticulum-associated mRNA-binding proteins ERBP1 and ERBP2 interact in bloodstream-form *Trypanosoma brucei*

Kathrin Bajak[1,2], Kevin Leiss[1], Christine E. Clayton[1] and Esteban Erben[2]

[1] Zentrum für Molekulare Biologie der Universität Heidelberg (ZMBH), Heidelberg, Germany
[2] Deutsches Krebsforschungszentrum (DKFZ), Heidelberg, Germany

## ABSTRACT

Kinetoplastids rely heavily on post-transcriptional mechanisms for control of gene expression, and on RNA-binding proteins that regulate mRNA splicing, translation and decay. *Trypanosoma brucei* ERBP1 (Tb927.10.14150) and ERBP2 (Tb927.9.9550) were previously identified as mRNA binding proteins that lack canonical RNA-binding domains. We show here that ERBP1 is associated with the endoplasmic reticulum, like ERBP2, and that the two proteins interact in vivo. Loss of ERBP1 from bloodstream-form *T. brucei* initially resulted in a growth defect but proliferation was restored after more prolonged cultivation. Pull-down analysis of tagged ERBP1 suggests that it preferentially binds to ribosomal protein mRNAs. The ERBP1 sequence resembles that of *Saccharomyces cerevisiae* Bfr1, which also localises to the endoplasmic reticulum and binds to ribosomal protein mRNAs. However, unlike Bfr1, ERBP1 does not bind to mRNAs encoding secreted proteins, and it is also not recruited to stress granules after starvation.

Corresponding author
Esteban Erben,
e.erben@dkfz-heidelberg.de

## INTRODUCTION

In *Trypanosoma brucei*, most regulation of gene expression is post-transcriptional. Protein-coding genes are arranged in polycistronic transcription units, and mRNAs are excised by *trans* splicing and polyadenylation (*Clayton, 2019*). Levels of constitutively expressed proteins and mRNAs are strongly influenced by codon usage (*De Freitas Nascimento et al., 2018*; *Jeacock, Faria & Horn, 2018*), while regulation during development, over the cell cycle, and in response to environmental conditions is effected mainly by RNA-binding proteins. The latter often, but not always, bind sequences in 3′-untranslated regions (3′-UTRs) (*Clayton, 2019*). All mRNAs—whether or not they are subject to specific regulation—are expected to be bound by numerous different proteins, forming a "messenger ribonucleoprotein" (mRNP) assembly.

*T. brucei* proliferates in mammalian blood and tissue fluids, and in the digestive system of Tsetse flies. The developmental stages that are most accessible to laboratory study are

the bloodstream form, which grows at 37 °C in glucose-rich media and corresponds to the form that grows in mammals, and the procyclic form, which is grown at 27 °C in proline-rich medium and multiplies in the Tsetse midgut. Purification of mRNPs from bloodstream forms revealed at least 155 proteins that reproducibly could be cross-linked to, and co-purified with, mRNA (*Lueong et al., 2016*). Although many of the mRNA-binding proteins had recognizable consensus RNA-binding motifs, the list included 49 proteins, including that encoded by Tb927.10.14150, which had no obvious connection to RNA metabolism. Similar studies of Opisthokonts also revealed numerous novel proteins without known RNA-binding domains, which have been named "enigmRBPs" (*Beckmann et al., 2015*; *Hentze et al., 2018*). We will, therefore, call the Tb927.10.14150 protein ERBP1 (for EnigmRBP1).

To assess the ability of trypanosome proteins to affect mRNA stability or translation, they were "tethered" to a reporter RNA. To do this, proteins or protein fragments were expressed fused to the N peptide from bacteriophage lambda. The lambdaN peptide binds an RNA stem-loop called boxB with high affinity. The reporter mRNA encoded a selectable marker with boxB sequences in the 3′-UTR. Proteins were screened first as random fragments (*Erben et al., 2014*), then at full length (*Lueong et al., 2016*). Numerous regulators were found, many of which were also in the mRNP proteome (*Lueong et al., 2016*), and ERBP1 reproducibly conferred a selective advantage when tethered both as fragments and at full length (*Erben et al., 2014*; *Lueong et al., 2016*). ERBP1 fused C-terminally to GFP was associated with the endoplasmic reticulum in a high-throughput screen of procyclic forms (*Dean, Sunter & Wheeler, 2016*) and its depletion resulted in a selective disadvantage in a high throughput RNAi screen (*Alsford, Glover & Horn, 2005*).

In this paper, we describe more detailed studies of ERBP1. ERBP1 is predicted to belong to a protein family that is named after a *S. cerevisiae* protein called Bfr1p (IPR039604 or PTHR31027), with alignment over its entire length. Bfr1p was originally recovered in a screen for high-copy-number suppression of Brefeldin A toxicity (*Jackson & Kepes, 1994*). It is an mRNA-binding protein that is associated with polysomes and the endoplasmic reticulum (ER) (*Lang et al., 2001*; *Weidner et al., 2014*), and is incorporated into stress granules (*Simpson et al., 2014*). In yeast, it is associated with over 1,000 different mRNAs, enriched for those encoding ribosomal proteins and mRNAs that are translated at the ER (*Lapointe et al., 2015*). In addition to a role in stress granules, Bfr1 has been implicated in ER quality control (*Low et al., 2014*) and correct nuclear segregation (*Xue et al., 1996*). The physical interaction map (https://www.yeastgenome.org/locus/S000005724/interaction) includes five proteins related to mRNA decay: with Xrn1p, Dcp2p, Scp160p, Puf3p, and Asc1p (an orthologue of RACK1 that inhibits translation). Our results reveal that ERBP1 has both similarities with, and differences from, Bfr1.

## MATERIALS AND METHODS

### DNA manipulation and trypanosomes

Lister 427 strain trypanosomes expressing the tet repressor were used for all experiments, and were cultivated and transfected as described previously (*Alibu et al., 2004*). All plasmids

and oligonucleotides are listed in Table S3. Expression from tetracycline-inducible promoters was induced with 100 ng/ml tetracycline, and all growth studies were performed in the absence of selecting antibiotics. Cultures that were used for RNA and protein analysis had a maximum density of $1.5 \times 10^6$/ml.

## Western blotting

$3–5 \times 10^6$ cells were collected per sample, resuspended in 6x Laemmli Buffer and heated at 95 °C for 10 min. The samples were subjected to SDS-PAGE gel electrophoresis using 10% polyacrylamide gels. The gels were then stained with SERVA blue G or blotted on a 0.45 μm nitrocellulose blotting membrane (Neolabs). To verify the protein transfer, the membrane was stained with Ponceau S (SERVA). The membrane was blocked with 5% milk in TBS-Tween and incubated with appropriate concentrations of first and secondary antibodies. Western Lightning Ultra (Perkin Elmer) was used as a chemiluminescence system and signals were detected with the LAS-4000 imager (GE Healthcare) and CCD camera (Fujifilm). Antibodies used were: rabbit anti-Aldolase (1:50000) (*Clayton, 1987*); mouse anti-myc 9E10 (Santa Cruz, 1:200); rabbit Peroxidase anti-Peroxidase (Sigma, 1:20000); rat anti-ribosomal protein S9 (1:1000); anti-Trypanothione Reductase (rabbit, gift from L. Krauth-Siegel, BZH Heidelberg); mouse anti-V5 (Biorad, 1:2000); anti-SCD6 and anti-DHH1 (from S. Kramer, University of Wurzburg, 1:10000 and 1:15000 respectively) and rabbit anti-BiP (from J. Bangs, University of Buffalo, 1:1000).

## Digitonin and stress granule fractionation

For each sample, $3 \times 10^7$ cells were collected by centrifugation at 2,000 g for 10 min at 4 °C. The pellet was resuspended in 100 μl 1x PBS and centrifuged at 10,000 g for 5 min at 4 °C. The pellet was then resuspended in 50 μl STE buffer (10 mM Tris–HCl pH 8.0, 150 mM NaCl, 1 mM EDTA) and centrifuged at 10,000 g for 5 min at 4 °C. A 10 μg/μl digitonin stock solution was heated at 98 °C for 5 min and cooled down before use. Seven different digitonin containing solutions, ranging from 0–1.65 μg/μl digitonin, were prepared and each pellet was resuspended properly in 60 μl of one solution. The samples were incubated at 25 °C for 5 min and then centrifuged immediately at 10,000 g and 4 °C for 5 min. The supernatant was transferred to another tube containing 20 μl 4x SDS-PAGE sample buffer. The pellet was washed twice with $1 \times$ PBS by centrifugation (4 °C, 10,000 g, 5 min) and finally resuspended in 80 μl $1 \times$ Laemmli buffer. Samples were analyzed by Western Blotting. Stress granules were purified exactly as described in *Fritz et al. (2015)*, and immunoprecipitations were done as described in *Singh et al. (2014)*.

## Immunofluorescence microscopy

Tissue culture glass slides with 8 chambers were treated with 0.1% Poly-Lysine (Sigma, P-8920). $2.5 \times 10^6$ formaldehyde-fixed *T. brucei* were allowed to adhere to poly-lysine-treated chambered slides (Falcon, 354108), permeabilised with 0.2% (w/v) Triton X-100 then incubated with protein-specific antibodies followed by fluorescently-labelled second antibodies in PBS containing 0.5% gelatin. DNA was stained with 100 ng/ml DAPI (D9542, Sigma-Aldrich). Mitochondria were detected by addition of Mitotracker Red CMXRos (50 nM, Thermo Fisher Scientific) to the cells 5 min prior to fixation. Images were examined

using the Olympus IX81 microscope, 100x Oil objective with a numerical aperture of 1.45. Digital images were taken with ORCA-R2 digital CCD camera C10600 (Hamamatsu) and using the Xcellence RT software. The bright-field images were taken using differential interference contrast (DIC). Fluorescent images were taken as Z-Stacks with a height of roughly 4 μm and a step width of 0.2 μm. The images were deconvoluted (Wiener Filter, Sub-Volume overlap: 20) and then processed using ImageJ. The background was subtracted and brightness and contrast were adjusted automatically. The most in-focus image of the deconvoluted stack was used.

## RNA preparation and Northern blotting

$5 \times 10^7$ cells were used for the extraction of total mRNA using peqGold Trifast (PeqLab). RNA was separated on agarose-formaldehyde gels and blotted on a nylon membrane (Amersham Hybond-N+, GE Healthcare, RPN203B). RNA was cross-linked on the membrane by UV light (0.2 J/cm$^2$) and stained afterwards with methylene blue (SERVA) before hybridization with $^{32}$P-labelled probes, made using either the Prime-IT RmT Random Primer Labelling Kit (Stratagene) or, for oligonucleotides, $[\gamma^{32}P]$ATP and T4 polynucleotide kinase (New England Biolabs).

## Affinity purification and mass spectrometry

To purify TAP-ERBP1 for mass spectrometry, the protein was subjected to two steps of affinity purification (*Estévez et al., 2003*). Briefly, the cleared lysate was incubated with IgG sepharose beads, washed, and then bound proteins were released using TEV protease. The resulting preparation was then allowed to adhere to a calmodulin affinity column, and proteins were eluted with EGTA. Co-purifying proteins from three independent experiments were analyzed by LC/MS by the ZMBH Mass Spectrometry facility. Cell lines expressing TAP-GFP served as control. Raw data were analyzed using MaxQuant 1.5.8.3, with label-free quantification (LFQ), match between runs (between triplicates), and the iBAQ algorithm enabled. The identified proteins were filtered for known contaminants and reverse hits, as well as hits without unique peptides.

## Affinity purification and RNASeq

For identification of associated RNAs (*Droll et al., 2013*), $1 \times 10^9$ cells were resuspended in 50 ml ice-cold PBS, and UV-crosslinked ($2 \times 0.24$ J/cm$^2$, Stratagene UV crosslinker) in two P15 Petri dishes on ice. They were then pelleted, snap-frozen and stored in liquid nitrogen before use. The TAP-ERBP1 was allowed to bind to IgG beads, and the unbound lysate was retained as one of the controls. The bound protein was released using TEV protease. Cross-linked proteins were removed by proteinase K digestion (*Droll et al., 2013*) and RNA was extracted using peqGold Trifast (peqLab) according to manufacturer's protocol. rRNA was depleted as required, by incubation with complementary oligonucleotides and RNaseH (*Minia et al., 2016*). The NEBNext Ultra RNA Library Prep Kit for Illumina (New England BioLab) was used for library preparation, prior to sequencing with sequenced at EMBL (HiSeq 2000) to generate 50-base reads. Data were analyzed using an in-house tool (10.5281/zenodo.165132) (*Mulindwa et al., 2018*).

## RESULTS

### ERBP1 is conserved in *Trypanosoma*

To investigate the function of ERBP1, we first analyzed its sequence. ERBP1 is a 55-kDa protein with an isoelectric point of 5.73. It is predicted (by Phyre2) to consist predominantly of alpha-helices. Homologues are found at the equivalent chromosomal position in other *Trypanosoma* species, and in *Endotrypanum*, *Paratrypanosoma*, *Blechomonas* and *Bodo*, but the gene appears to have been lost in *Leishmania*. The alignment (Fig. S1) shows that the proteins share extremely acidic C-termini: for example, in *T. brucei*, the last 33 amino acid residues include 5 aspartates and 14 glutamates, as opposed to 3 lysines. Outside the Kinetoplastida, the nearest matches were not with yeast Bfr1p, but with proteins of unknown function in *Galdieria sulphuraria*, an acidiphilic red alga, and organisms from the SAR group that probably have incorporated red algal endosymbionts: the oomycete *Phytophthora parasitica*, the brown alga *Ectocarpus siliculosus* and the diatom *Phaeodactylum tricornutum*. The alignment with all these proteins, as well as Bfr1p, showed little sequence identity between the kinetoplastid sequences and all the others (Fig. S1).

### ERBP1 is required for normal growth but is not essential

We first confirmed that ERBP1 is indeed required for normal growth of bloodstream-form trypanosomes. To do this, we created bloodstream forms in which one *ERBP1* gene was modified to encode a protein with an N-terminal V5 tag (V5-ERBP1). These were transfected with a tetracycline-inducible construct for production of *ERBP1* dsRNA. Depletion of ERBP1 clearly inhibited cell proliferation but the cells nevertheless survived (Fig. 1).

Since some protein always remains after RNAi, we also tested the ability of the cells to survive in the absence of the protein. Bloodstream form cells without ERBP1 (Figs. 2A and 2B) also grew slower than wild-type (Fig. 2C), and this could be compensated by re-expression of the protein (Fig. 2D). Cells containing only a single tetracycline-inducible copy of ERBP1-myc (inducible on a knock-out background, cKO) grew at similar rates to wild-type with or without tetracycline, but the protein was clearly detectable in the absence of induction. The double band from ERBP1 was not always seen but might be caused by acetylation (*Moretti et al., 2018*) or phosphorylation (*Urbaniak, Martin & Ferguson, 2013*). Despite these initial results, upon prolonged cultivation, the cells lacking ERBP1 gradually increased their growth rate to wild-type. Survival and recovery of the cells after starvation was also indistinguishable from wild-type (Fig. S2A). In procyclic forms, the endogenous ERBP1 gene could be deleted only if an inducible ERBP1-myc gene was present. Depletion of the ERBP1-myc had no effect on growth or recovery from starvation (Fig. S2B), but some residual ERBP1-myc was doubtless present. It is, therefore, possible that ERBP1 is essential in procyclic forms, but the failure of the deletion could also have been due to a technical problem.

The other high-throughput result that required verification was the effect of ERBP1 when tethered to a reporter mRNA. Full-length ERBP1 had previously been identified as conferring a selective advantage in the screen, suggesting that it could activate expression of the blasticidin resistance marker. It gave no growth advantage in control cells in

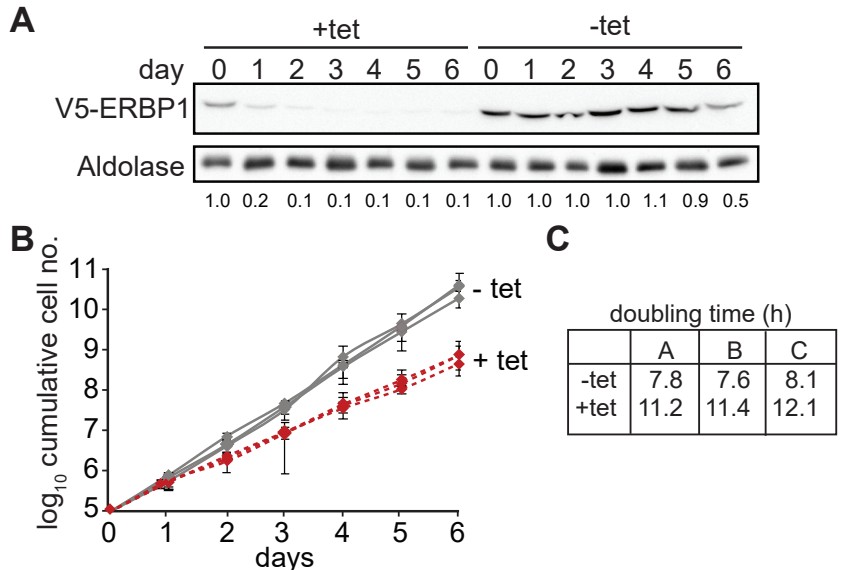

**Figure 1** **ERBP1 is required for normal growth of bloodstream-form *T. brucei*.** Bloodstream forms in which one *ERBP1* gene was V5-tagged *in situ* at the 5′-end were used, with inducible expression of stem-loop *ERBP1* dsRNA. (A) Depletion of V5-ERBP1 after RNAi. $3 \times 10^6$ cells were collected each day and V5-ERBP1 was detected by Western blotting. Aldolase was used as loading control. Numbers below the blot indicate the quantification of the V5-ERBP1 signals normalized to the uninduced RNAi at day 0. (B) Growth curve for three Tet+ (red) and Tet- (grey) clones analysed in three independent experiments. (C) Doubling times (in h) for the 3 experiments are shown.

which the marker mRNA had no boxB sequences, ruling out the possibility that ERBP1 expression by itself results in blasticidin resistance. However, lambdaN-ERBP1 was unable to increase expression of a chloramphenicol acetyltransferase reporter (Fig. S2A). We have no explanation of this discrepancy: for both the screen and the individual test, the lambdaN-myc sequence was placed at the N-terminus of the open reading frame.

## ERBP1 is associated with the endoplasmic reticulum and with a second ER protein

To examine the location of ERBP1 in bloodstream-form cells, we examined either N-terminally V5-tagged ERBP1 (V5-ERBP1), or C-terminally myc-tagged ERBP1 (ERBP1-myc) expressed from the endogenous locus (Figs. 3 and 4). In both cases, ERBP1 clearly co-localized with the endoplasmic reticulum and to some extent could be found in close proximity to mitochondria and glycosome. We then conducted controlled-digitonin permeabilization studies to evaluate the release of the protein with the same markers (Fig. 5). ERBP1 was found to coelute partially with the cytosolic marker but fully with the endoplasmic reticulum marker, suggesting that it is loosely associated with the cytosolic face of the ER.

To investigate the interactions of ERBP1, we integrated a sequence encoding the tandem affinity purification (TAP) tag into the genome such that the tag would be at the ERBP1 N-terminus (TAP-ERBP1) while the other copy of *ERBP1* was deleted. TAP-tagged GFP

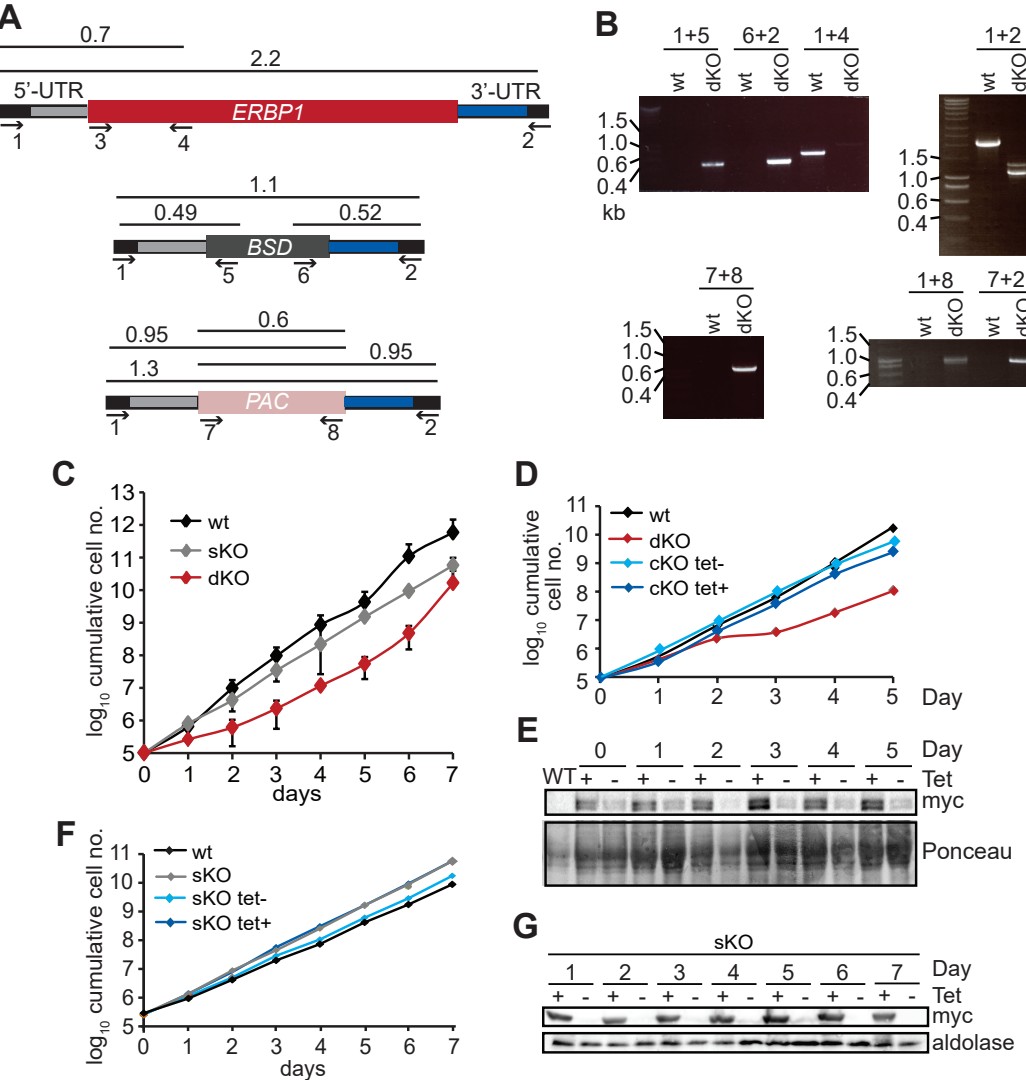

**Figure 2** **Bloodstream-form trypanosomes survive without ERBP1.** (A) Schematic representation of the *ERBP1* gene before and after replacement with selectable marker genes. Primers for PCR and corresponding product lengths (in kb) are shown. (B) DNA from wild type (wt) and double knock-out (dKO) cells was analysed by PCR using the primers shown in (A). The primers used are indicated above each image. (C) Cumulative growth curve for 3 independent experiments, each with three different clones for wild-type (wt), single knock-out (sKO) and double knock-out (dKO). Error bars indicate standard deviation from results for 3 clones; each was measured 3 times then for each, and average was taken. (D) Expression of ERBP1-myc complements the defect in dKO cells. The dKO cells were complemented with tetracycline-inducible ERBP1-myc (complemented KO; cKO). The graph shows the cumulative parasite numbers. (E) Western blot from (D) using $3 \times 10^6$ cells/lane. Some ERBP1-myc is detectable in the absence of tetra-cycline inducer. (F) As (D) but for procyclic forms. As homozygous gene replacement failed in procyclic forms, the sKO cells were complemented with tetracycline-inducible ERBP1-myc. (G) Western blot from (F) using $3 \times 10^6$ cells/lane. Aldolase was used as loading control.

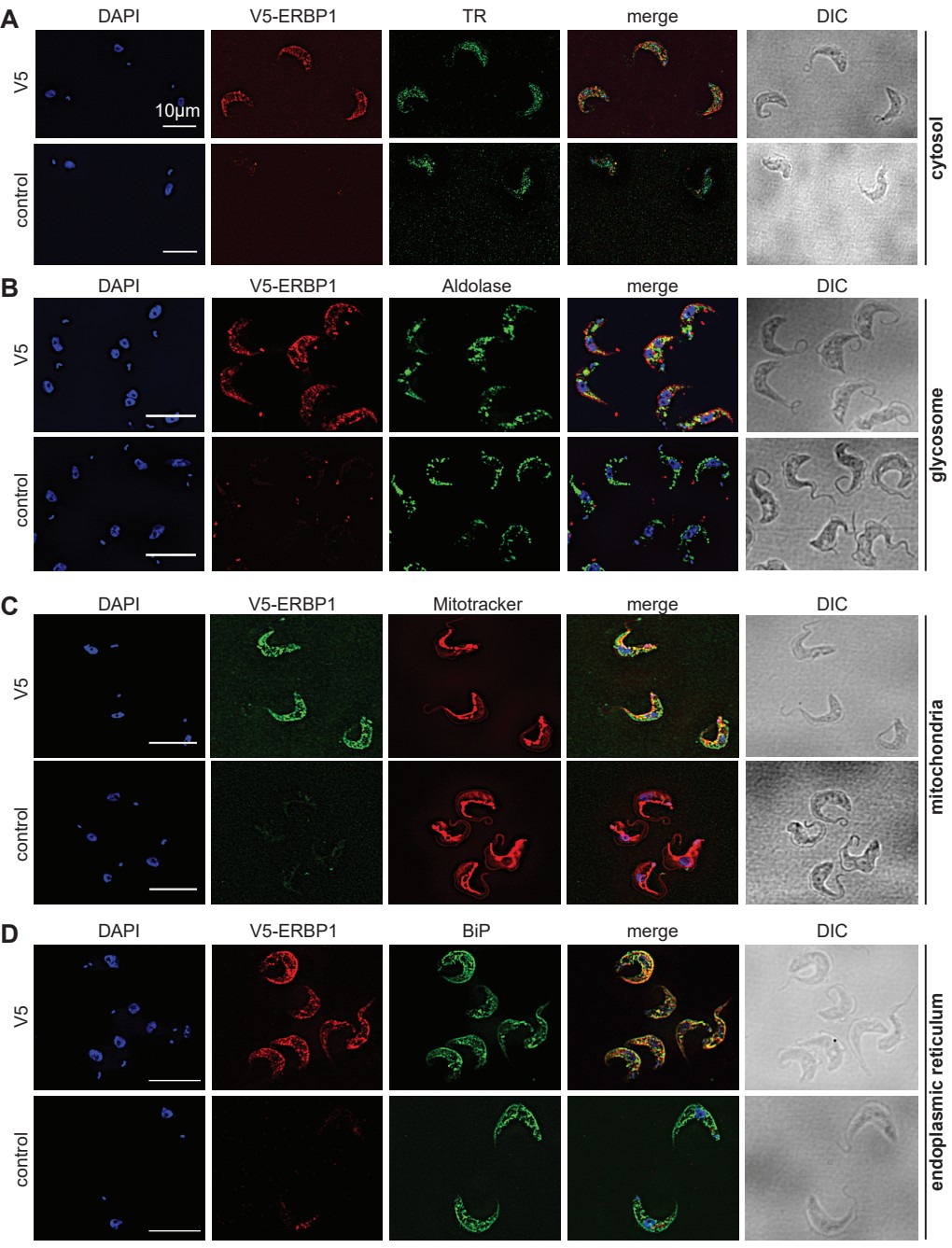

**Figure 3** **V5-ERBP1 colocalises with the endoplasmic reticulum.** Bloodstream forms expressing N-terminally V5-tagged ERBP from the endogenous locus (V5-ERBP1) were examined by immunofluorescence microscopy. Cells without V5 served as control. Nuclear and kinetoplast DNA was stained with DAPI (blue). Representative images are shown for three independent experiments. Z-stacks were examined using the Olympus CellR microscope and 100 × magnification, and images were deconvoluted. Scale bar: 10 μm. (A) V5-ERBP1 is red and cytosolic Trypanothione Reductase (=TR) is green. (B) V5-ERBP1 is red and glycosomal aldolase is green. (C) V5-ERBP1 is green and mitotracker is red. (D) V5-ERBP1 is red and the endoplasmic reticulum marker BiP is green.

served as the control. Three proteins reproducibly co-purified with ERBP1: calmodulin, the ATP-dependent RNA helicase HEL67/Vasa-like (*Kramer et al., 2012*), and a protein of unknown function encoded by Tb927.9.9550 (Table S1). Calmodulin quite often co-purifies with TAP-tagged proteins (*Schimanski, Nguyen & Günzl, 2005*). The association with the RNA helicase may be significant; although we have detected it on many other mRNA-related purifications. Tb927.9.9550, however, was a novel protein partner. It is conserved throughout Kinetoplastida, activating in the tethering screen (*Erben et al., 2014*; *Lueong et al., 2016*), is in the mRNP proteome (*Lueong et al., 2016*), but again lacks any recognizable domains; we, therefore, call it ERBP2. It appeared to be essential in the high-throughput RNAi screen (*Alsford, Glover & Horn, 2005*), has a predicted signal peptide and trans-membrane domain, and a GFP-tagged version, like ERBP1, localized to the ER in procyclic form (*Dean, Sunter & Wheeler, 2016*). To verify the interaction we used bloodstream-form cells containing YFP-*in-situ* tagged ERBP1, with or without expression of ERBP2-myc. YFP-ERBP1 was pulled down with anti-myc antiserum only if ERBP2-myc was present, confirming the interaction (Fig. 6A), but the experiment also showed that—at least under the conditions used—only a very small proportion of the ERBP1 was associated with ERBP2. This may, therefore, be a transient interaction. ERBP2 is predicted to be predominantly alpha-helical, like ERBP1. The isoelectric point of ERBP2 is 11, so it might interact with the highly acidic ERBP1 C-terminus.

## ERBP1 associates with mRNAs encoding ribosomal proteins, and not with starvation stress granules

Finally, we wished to know whether ERBP1 had any mRNA-binding specificity. We, therefore, sequenced mRNAs that were co-purified with TAP-ERBP1 from bloodstream-form cells, comparing them either with one unbound fraction or with the total mRNA. Unusually, the *ERBP1* mRNA itself was not enriched, as would be expected from pull-down of the nascent polypeptide. However, the two bound fractions clearly separated from all the controls (Fig. 6B). Sixty four mRNAs were enriched at least 2-fold in both pull-downs relative to all controls. Strikingly, nearly half of them (29) encode ribosomal proteins. Two glycosomal membrane protein mRNAs, PMP4 and PEX11, were also enriched, but those encoding other PEX proteins—including other glycosomal membrane proteins—were not, so the significance of this is uncertain. The length of the associated mRNAs seemed to be important since both ribosomal and non-ribosomal targets are shorter than the median (not shown).

It should be noted that the enrichment of mRNAs is calculated relative to the enrichment of other mRNAs, and does not actually tell us which proportion of mRNAs with that sequence is associated with the protein *in vivo*.

In budding yeast, Bfr1 is associated with stress granules. To find out whether ERBP1 is stress-granule associated, we incubated procyclic forms in PBS for 2 h. Although clear granules containing the marker SCD6 were formed (Fig. 7A), V5-ERBP1 remained distributed throughout the cytosol (Fig. 7A) and was not associated with the granule fraction (Fig. 7B). There is, therefore, no evidence that ERBP1 is involved in the formation of stress granules or survival after starvation.

 

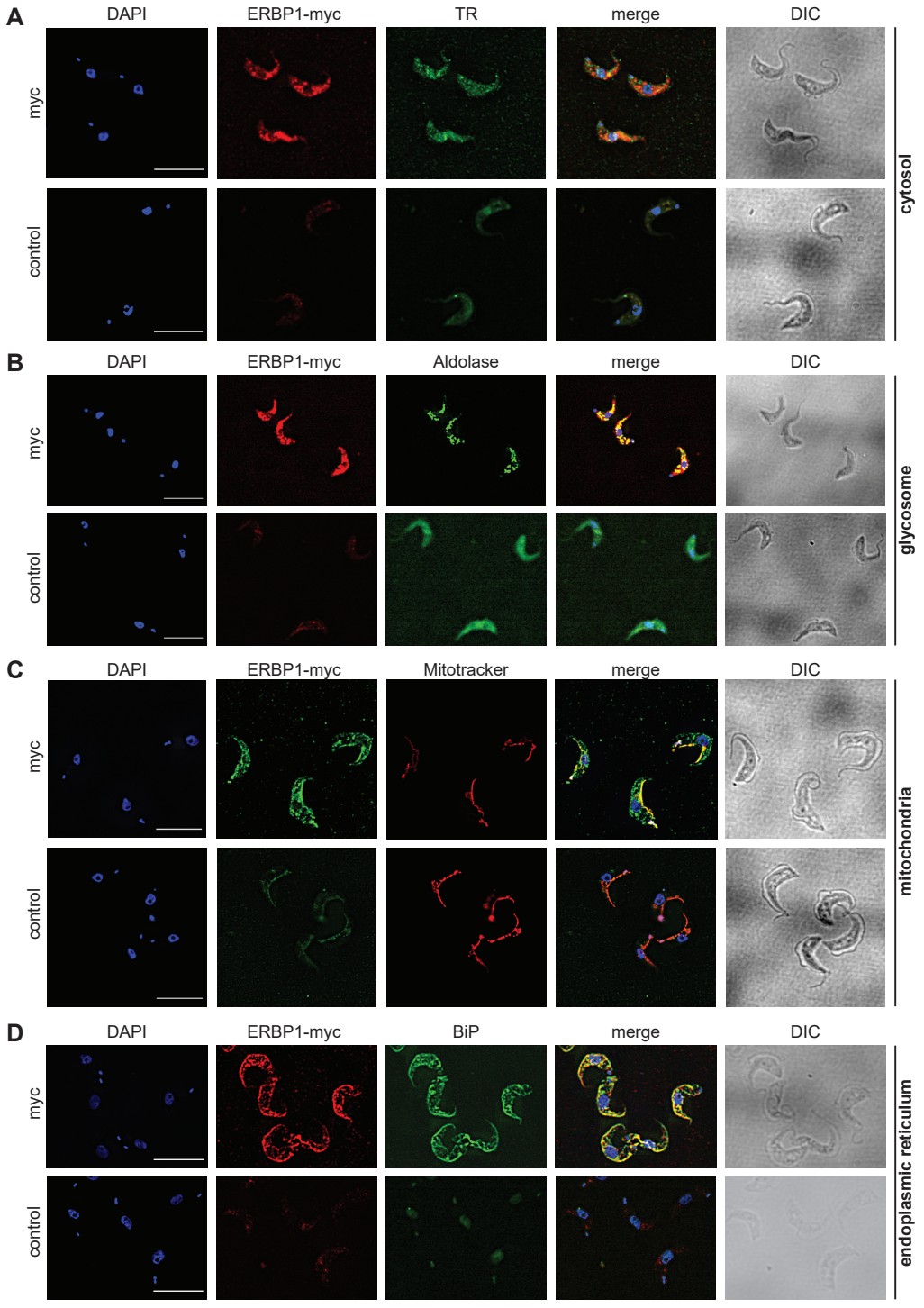

**Figure 4 ERBP1-myc colocalises with the endoplasmic reticulum.** Bloodstream forms expressing C-terminally myc-tagged ERBP from the endogenous locus (ERBP1-myc) were examined by immunofluorescence microscopy. Cells without myc served as control. Nuclear and 

kinetoplast DNA was stained with DAPI (blue). Representative images are shown for three independent experiments. Z-stacks were examined using the Olympus CellR microscope and 100 × magnification, and Images were deconvoluted. Scale bar: 10 μm. (A) ERBP1-myc is red and cytosolic Trypanothione Reductase (TR) is green. (B) ERBP1-myc is red and glycosomal aldolase is green. (C) ERBP1-myc is green and mitotracker is red (D) ERBP1-myc is red and the endoplasmic reticulum marker BiP is green.

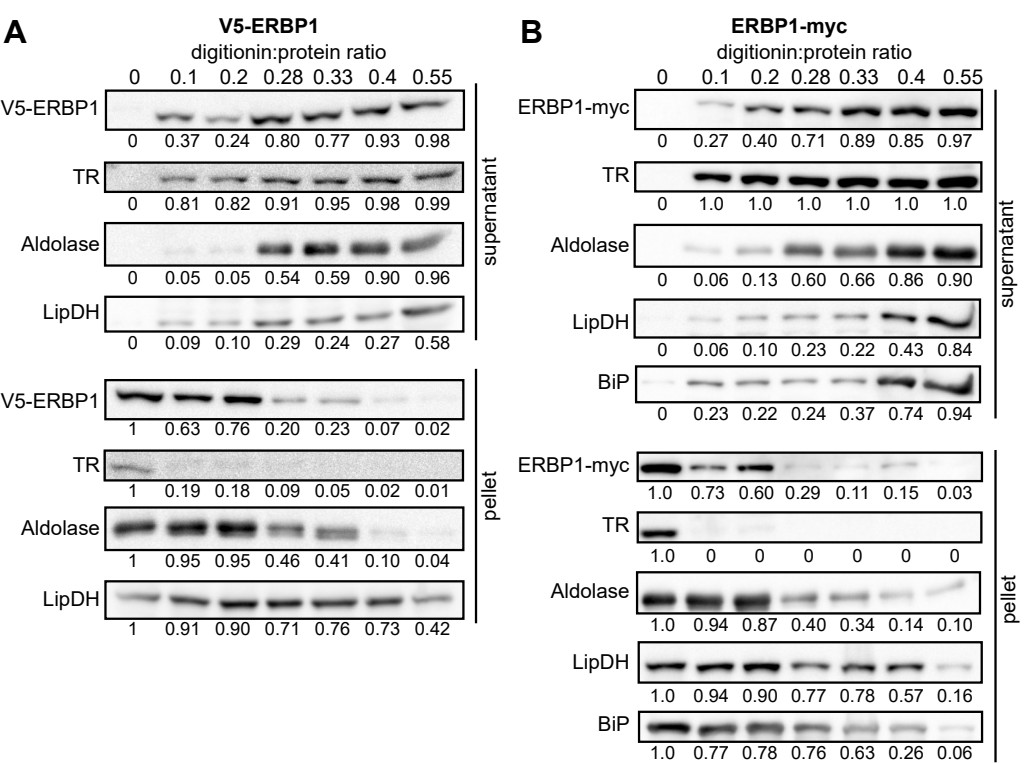

**Figure 5** **ERBP1 is not inside a membrane-bound compartment.** Bloodstream-form cells expressing V5-ERBP1 (A) or ERBP1-myc (B) were treated with increasing digitonin:protein (mg:mg) ratios as indicated at the top. Samples were centrifuged and the supernatant and pellet were analysed by Western blotting. Numbers below the blots indicate quantification of the corresponding signal. Aldolase (glycosome); TR: Trypanothione Reductase (cytosol); LipDH: Lipoamide Dehydrogenase (mitochondria); BiP: Binding Immunoglobulin Protein (ER).

## DISCUSSION

Our results suggest that ERBP1 and its partner ERBP2 may modulate ribosomal expression, and are both associated with the endoplasmic reticulum. Our results revealed both similarities and differences between ERBP1 and Bfr1. Each is associated with the endoplasmic reticulum (*Lang et al., 2001*), and each binds preferentially to mRNAs encoding ribosomal proteins (*Lapointe et al., 2015*). ERBP1, however, unlike Bfr1, does not show any preference for mRNAs encoding proteins that are imported into the endoplasmic reticulum, and it does not associate with stress granules. The latter observation is consistent with the exclusion of ribosomal protein mRNAs from trypanosome stress granules (*Fritz*

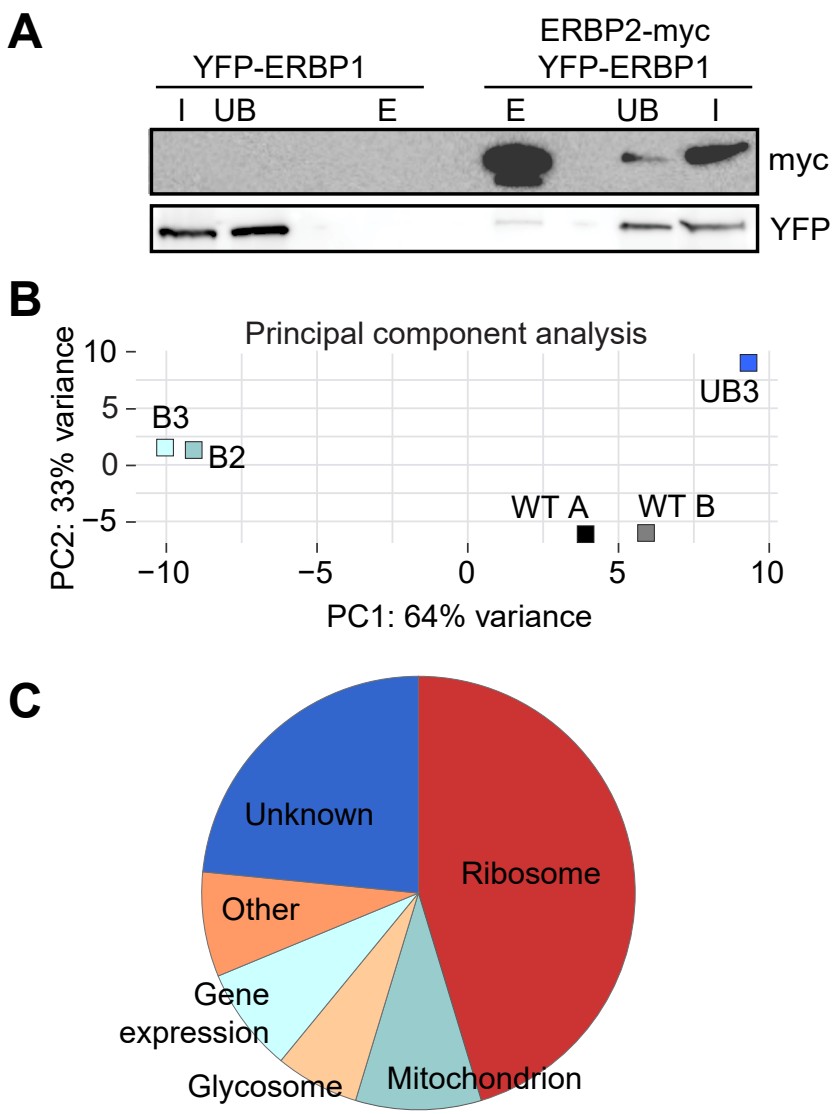

**Figure 6** **Interactions of ERBP1.** (A) ERBP2 co-precipitates with ERBP1. Extracts were made from cells expressing YFP-ERBP1, with or without ERBP2-myc. After immunoprecipitation with anti-myc, a Western blot was probed for myc and for YFP as indicated. I, input; UB, unbound fraction; E, eluate from anti-myc immunoprecipitation. (B) Principal component analysis of transcriptomes after RNA-Seq of total mRNA (WT) and after pull-down of TAP-ERBP1; UB, unbound; B, bound (eluate from the affinity matrix). (C) Functions of proteins encoded by ERBP1-associated mRNAs. The pie chart show the distributions for mRNAs that were at least 2-fold enriched in both bound fractions relative to the controls. Details are in Table S2.

*et al., 2015*). ERBP1 is also associated with a kinetoplastid-specific protein, ERBP2 which, like ERBP1, was detected in the mRNP proteome (*Lueong et al., 2016*).

Synthesis of ribosomes occupies considerable resources in all growing cells, which leads to a requirement for coordination between ribosomal protein synthesis and rRNA transcription. In mammalian cells and yeast, this coordination happens in the nucleus, at the level of transcription (*Albert et al., 2016*; *Calo et al., 2015*). Trypanosomes also devote

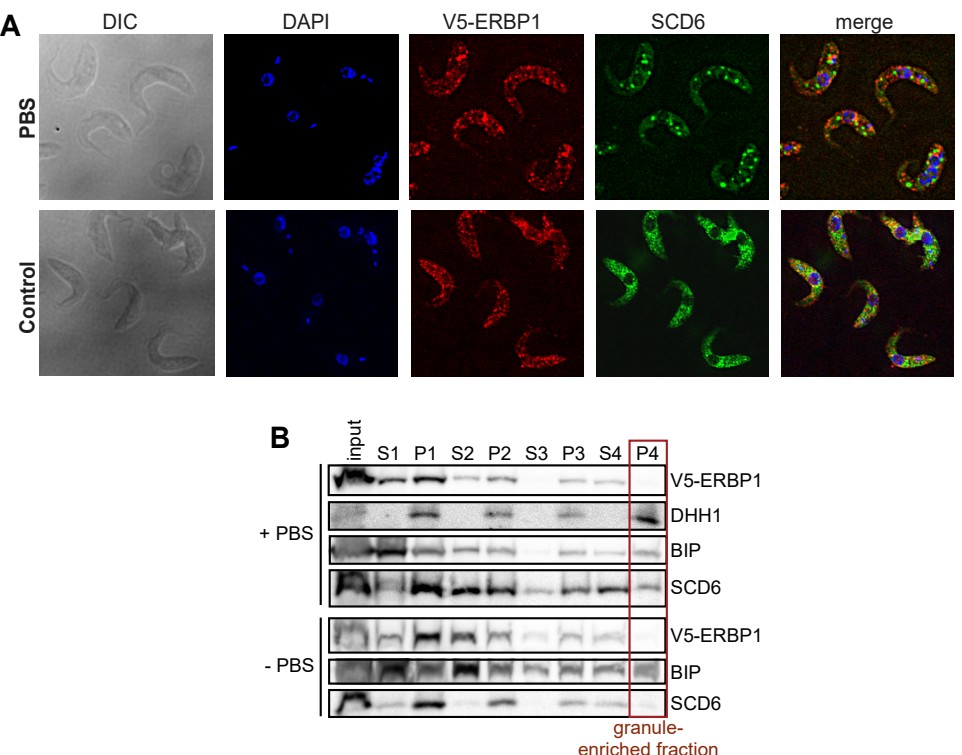

**Figure 7  V5-ERBP1 is not concentrated in starvation stress granules.** (A) Procyclic forms expressing V5-ERBP1 were incubated for 2 h in 1 × PBS, then fixed and stained for V5 (red) and the stress granule marker SCD6 (green). Nuclear and kinetoplast DNA were stained with DAPI (blue). Z-stacks were examined using the Olympus CellR microscope and 100 × magnification. Images were deconvoluted. The control was not starved in PBS. (B) V5-ERBP1-expressing procyclic forms were starved as in (A) and then fractionated to concentrate stress granules.

considerable resources to ribosome synthesis: in bloodstream-forms, resources devoted to rRNA transcription are at least as great as those needed for the synthesis of mRNAs (*Haanstra et al., 2008*), and the ribosomal protein mRNAs together constitute more than 10% of total mRNA (*Fadda et al., 2014*). Feedback inhibition of rRNA processing is known to occur in trypanosomes upon disruption of ribosome assembly or export (*Droll et al., 2010*), but coordination of rRNA synthesis with ribosomal protein availability—or vice-versa—has not been investigated. The mRNAs encoding trypanosome ribosomal proteins are longer-lived than most others (*Fadda et al., 2014*), so at least in the short term, control of their translation might be required in order to respond to altered conditions. Ribosome densities on ribosomal protein mRNAs are significantly lower than on other mRNAs, and most of the open reading frames are also relatively short (median 0.6 kb); consequently on average there is only one ribosome per mRNA (*Antwi et al., 2016*). Low ribosome densities occur if the rate of peptide chain elongation is fast relative to the rate of translation initiation. Ribosomal protein mRNAs indeed have optimal codon usage (*De Freitas Nascimento et al., 2018*), which would result in rapid elongation, and their short 5′-untranslated regions (*Clayton, 2019*) might assist scanning after 43S complex

recruitment. These characteristics would lead to high constitutive expression, but would also enable very rapid responses to altered conditions since, at 5–10 residues per second, the single ribosome would run off in less than 1 min. We, therefore, suggest that several redundant pathways control ribosome protein synthesis, and one of them includes ERBP1.

## CONCLUSIONS

In *Trypanosoma brucei*, ribosomal protein encoding mRNAs seem to be excluded from starvation stress granules; however, how granule exclusion of these mRNAs occurs is not known. Our studies suggest that ERBP1 is an RNA-binding protein that associates with ribosomal mRNAs, interacts with ERBP2, localizes to the cytosolic face of the endoplasmic reticulum and it does not associate with stress granules. If ERBP1 is the trans-acting factor needed for the exclusion of these mRNAs from starvation stress granules is still not known. As ERBP1 is required for normal growth but is not essential, we suggest that redundant pathways control ribosome synthesis, and one of them may include ERBP1.

## ACKNOWLEDGEMENTS

We thank Nina Papavisiliou for generously hosting KB during the latter part of this project. We also acknowledge Claudia Helbig and Ute Leibfried for technical support, David Ibberson of the BioQuant sequencing facility for cDNA library construction, and the Mass spectrometry facility of the ZMBH. We thank Luise Krauth-Siegel, (BZH Heidelberg), Susanne Kramer (University of Würzberg) and Jay Bangs (University of Buffalo) for antibodies, and Pia Hartwig for work done during a lab rotation.

### Funding

This project was partially supported by the Deutsche Forschungsgemeinschaft (grant Cl112/24 to CC). The funders had no role in study design, data collection and analysis, decision to publish, or preparation of the manuscript.

### Grant Disclosures

The following grant information was disclosed by the authors:
Deutsche Forschungsgemeinschaft: Cl112/24 to CC.

### Competing Interests

Christine Clayton is an Academic Editor for PeerJ.

### Author Contributions

- Kathrin Bajak conceived and designed the experiments, performed the experiments, analyzed the data, prepared figures and/or tables, authored or reviewed drafts of the paper, and approved the final draft.
- Kevin Leiss analyzed the data, prepared figures and/or tables, and approved the final draft.

- Christine E. Clayton conceived and designed the experiments, analyzed the data, prepared figures and/or tables, authored or reviewed drafts of the paper, and approved the final draft.
- Esteban Erben conceived and designed the experiments, analyzed the data, authored or reviewed drafts of the paper, and approved the final draft.

## Data Availability

RIPseq data is available at ArrayExpress: E-MTAB-8257. Mass spectrometry data is available at Pride: PXD016136. Full-length uncropped blots are available at Figshare: Erben, Esteban (2019): ERBP1 original data. figshare. Figure. 10.6084/m9.figshare.9758792.v1.
https://www.ebi.ac.uk/arrayexpress/experiments/E-MTAB-8257
https://www.ebi.ac.uk/pride/archive/projects/PXD016106.

## Supplemental Information

Supplemental information for this article can be found online at http://dx.doi.org/10.7717/peerj.8388#supplemental-information.

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
