# Peer review of "The endoplasmic reticulum-associated mRNA-binding proteins ERBP1 and ERBP2 interact in bloodstream-form Trypanosoma brucei"

_PeerJ, doi:10.7717/peerj.8388_

## Round 0.1 · original submission · Minor Revisions

I apologize for the slow response to your submission. This was mainly because of the problems to find a second reviewer (the first review was submitted long ago), as, for some reason, several potential reviewers could not be reached or rejected the invitation. But I hope the decision will be rewarding enough because the reviewers agreed on a good level of the paper. Still, there are quite some number of minor revisions that are recommended before I could make a final decision. I would be glad to receive duly revised version of the manuscript addressing all the raised concerns soon.

·

Basic reporting

In this work, the authors characterised the trypanosome protein ERBP1, which they had previously identified as a potential regulator of mRNA levels in a high-throughput screen. The authors could not confirm the result of this initial screen when they tethered ERBP1 to a reporter RNA, but continued the characterisation of this non-conventional RNA binding protein. They found that the protein is not essential, at least not in the blood stage, but depletion or deletion decreased the growth rate. They show that the protein co-localises with the ER and that it preferentially binds mRNAs encoding ribosomal proteins. They identify one novel binding partner of ERBP1. Overall, this work is of interest to the trypanosome community and most of the data are in sufficient quality and convincing. The manuscript and the figures could have been prepared a little more carefully.

My comments

1) I could not find legends for the Sup Figures. I could therefore not fully judge these figures, in particular S3.

2) The mass spec and RNA seq data need to be submitted to online databases.

3) Please mention the life cycle stage for each chapter/experiment in the text.

4) Once you have a PubMedID, please update the comments on TriTrypDB, in particular, the contradictory results about the tethering experiments.

Figure 1
I don’t see any error bars. I don’t see data of nine experiments. Legend does not correspond to the figure (e.g. –TET is not red). Please correct figure and legend carefully.

Figure 2
The authors use three different nomenclatures, DKO, HKO and ERBP1-/- to name the ERBP1 knock-out cell line, this is confusing, why not stick to just ERBP1-/- (and +/- for the heterozygote). DKO is not even explained. HKO is in particular confusing as it could mean both homo and heterozygous knock out. Doubling time or population doubling time is more accurate then division time. For the procyclic cells, the cell line is labelled HKO in Figure 2E and Figure S2, but this should be heterozygote. Why does SKO grows better than WT?

line 94:
‘Since tetracycline-inducible expression is effected from a very active RNA polymerase I promoter, the result suggests that excess ERBP1-myc is 
not toxic. ‘ Without measuring the protein levels, nothing should be concluded as to whether overexpression is toxic or not. A promoter is only one component of many that determines protein levels. Best to omit this sentence.

Figure 5
The localisation of aldolase is not mentioned in the legend (although you mention before in Figure 3). I do not understand the quantifications underneath the bands, looks like percentages, but then, it goes up to 200. I do not agree to the conclusion from this figure. In B supernatant, ERBP1 looks clearly different from the cytoplasm to me, and not very different from the ER.

Why not load pellet and supernatant of each fractionation next to each other to the same gel and set both bands as 100%? This way, the data would become more comparable. The experiment should be done in triplicates. A graphic can then show averages of either supernatant or pellet with statistics.

Discussion

Line 161: ‘The latter observation is consistent with the exclusion of ribosomal protein mRNAs from trypanosome stress granules. ‘
It would be good to cite.


Ribosomal mRNAs tend to be very small. Is this true for the other associated RNAs too, could size be what matters for ERBP interaction?


Typos etc:

Figure legend for Figure 1: 3x106 (superscript 6)

Line 106: ‘It gave no growth advantage in control cells in which when the marker mRNA had no boxB sequences, ruling out the possibility that ERBP1 expression by itself results in blasticidin resistance.

Line 118: ‘To investigate the interactions of ERBP1, we used trypanosomes in which ERBP1 a sequence encoding a tandem affinity purification (TAP) tag was integrated at the N-terminus, and the other copy of ERBP1 was deleted.’


Line 123: ‘The association with the DED1 helicase may be significant, although we have identified on most other mRNA-related purifications.‘


Typos in Table S2 (legend), purified


Susanne Kramer

Experimental design

all covered in the above

Validity of the findings

all covered in the above

Additional comments

all covered in the above

Reviewer 2 ·

Basic reporting

A few places in which the English could be improved.

Abstract: ‘Results from a pull-down’ to Measurements from a puill-down’

Line 33: ‘T brucei grows’ to T brucei proliferates’.
It also proliferates in the tsetse salivary gland.\

Everything else is fine

Experimental design

no comment

Validity of the findings

Line 89:
‘Since some protein always remains after RNAi’
Undoubtedly true but given that the 10 000 fold increase in cell number of the induced RNAi (Figure 1B) how much and where did in come from?

Line 122:
Is there a citation for calmodulin contaminating pull downs?

---

## Round 0.2 · Minor Revisions

Thanks for following carefully the reviewers' recommendations and for submitting the advanced version of the paper. The revised manuscript still contained several minor inaccuracies, and I have found an opportunity to suggest my editorial corrections (I attach the new PDF file with track changes). Some of them are obviously your typos and small mistakes, but the list is by no means full, so please doublecheck carefully the entire final version of your paper before re-submitting, at which point we will be able to Accept it.

---

## Round 0.3 · accepted · Accept

There is no additional requests from my side.